# Are the Parameters of Novel Two-Point Force-Velocity Model Generalizable in Leg Muscles?

**DOI:** 10.3390/ijerph18031032

**Published:** 2021-01-25

**Authors:** Saša Đurić, Vladimir Grbić, Milena Živković, Nikola Majstorović, Vedrana Sember

**Affiliations:** 1Faculty of Sports, University of Ljubljana, 1000 Ljubljana, Slovenia; sasa.djuric@fsp.uni-lj.si; 2Faculty of Sport and Physical Education, University of Belgrade, 1000 Ljubljana, Slovenia; zver70@gmail.com (V.G.); milena.zix@gmail.com (M.Ž.); nikola.majstorovic@fsfv.bg.ac.rs (N.M.)

**Keywords:** exercise, resistance, performance, biomechanics, two-point model, force-velocity relationship

## Abstract

The two-point force-velocity model allows the assessment of the muscle mechanical capacities in fast, almost fatigue-free conditions. The aim of this study was to investigate the concurrent validity of the two-point parameters with directly measured force and power and to examine the generalization of the two-point parameters across the different functional movement tests of leg muscles. Twelve physically active participants were tested performing three functional lower limb maximal tests under two different magnitudes of loads: countermovement jumps, maximal cycling sprint, and maximal force under isokinetic conditions of the knee extensors. The results showed that all values from the two-point model were higher than the values from the standard tests (*p* < 0.05). We also found strong correlations between the same variables from different tests (r ≥ 0.84; *p* < 0.01), except for force in maximal cycling sprint, where it was low and negligible (r = −0.24). The results regarding our second aim showed that the correlation coefficients between the same two-point parameters of different lower limb tests ranged from moderate to strong (r −0.47 to 0.72). In particular, the relationships were stronger between power variables than between force variables and somewhat stronger between standard tests and two-point parameters. We can conclude that mechanical capacities of the leg muscles can be partially generalized between different functional tests.

## 1. Introduction

Muscle mechanical properties and their evaluation are known to be complex [1], as muscle strength depends on the current level of neural excitation, muscle contraction and time elapsed since the change in muscle excitation [1,2]. The slower a skeletal muscle shortens, the more force is generated during contraction (also force-velocity relationship), [3,4] is a basic principle of skeletal physiology [4]. Studies have been carried out first on isolated muscles and later on single and multi-joint movements [5]. Nevertheless, the expansion of scientific knowledge about the force-velocity relationship (F-V) began several years ago with the study of Jaric in 2015 [1], who proposed that F-V follows a linear form in multi-joint movements [5].

The standard testing procedures applied for the assessment of leg muscle capacities often consider the performance of a single external testing load [6,7] and therefore assessment in a single mechanical condition. Consequently, the outcomes observed in this way do not allow differentiation among different muscle capacities, such as those for generating high force (F), velocity (V) and power (P) [1,8]. In addition, standard testing procedures often include movements that are not specific to sports or daily activities [9], or they may cause excessive strain on the musculoskeletal system. The outcomes of most routine testing procedures have been of limited informational value and therefore a number of issues in research have originated from arbitrarily interpreted experimental findings on specific muscle capacities. 

As opposed to standard testing procedures, the F-V relationship of multi-joint movements provides the possibility to selectively assess F, V and P generating capacity of the tested muscles [1]. Although it has been accepted for several decades that the F-V relationship has an approximately hyperbolic shape [4,10], recent studies indicate that the proposed relationship appears to be approximately linear and strong for multi-joint movements [11,12,13,14,15]. Moreover, although it is considered curvilinear, the F-V relationship also appears to be linear for single joint movements, as tested by isokinetic dynamometry [10,16]. Several authors [9,13,17,18,19] have already suggested that the linear F-V relationship could be developed into a routine test of mechanical muscle capacity in elite sports [9,16] or in older adults [20]. It has been shown that the application of different loads (regression F-V model) is time consuming, prolongs the procedure and tends to cause fatigue [9,10,21]. These findings dictate the use of a recently proposed two-point (i.e., two-load) method for testing various movement tasks that involve only two different external loads [8,9,15,21]. Specifically, this method provides the parameters representing F0 (i.e., the force intercept), V0 (velocity intercept) and P0 (calculated from the product of F and V) of the tested muscles [16]. The two-point model allows the assessment of the muscle mechanical capacities in fast, almost fatigue-free conditions. Therefore, it is suitable for testing more sensitive populations, such as young athletes, professional athletes recovering from injury or the elderly [21,22]. These two-point parameters correspond to the standard linear F-V relationship parameters obtained from several external load magnitudes [15,23]. Therefore, adding an additional load to the standard tests could allow the assessment of the mechanical muscle capacities (i.e., F, V and P), providing a deeper insight into the function of the tested muscles and resolve a number of questions questioned in the literature. In addition, such knowledge could also improve the outcomes of muscle testing in different environmental scenarios and physiological conditions to understand the human body’s adaptations and reactions to temperature [24,25,26], altitude [27] or dehydration [28,29,30].

Although jumping on force platforms [31,32,33], cycling [32] and isokinetic dynamometry [23] are valid standard tests of leg muscle capacities and in assessing the F-V relationship [34], there is a lack of data regarding the relationship between outcomes of these tests. It should be kept in mind that the implicit assumption of any standard muscle capacity test is that the results typically observed in very few tests and muscles can be partially generalized to other muscle systems that perform different functional movements [33,35]. Accordingly, only one study examined the generalizability of the linear parameters of the F-V relationship for leg muscle capacities [15]. The authors concluded that the linear F-V relationship parameters could only partially be generalized to different muscle groups [15]. However, to our knowledge, the relationship and generalization between parameters obtained from a two-point model for leg muscle capacities have not yet been evaluated.

To address the issues discussed, we designed a study to investigate the two-point model parameters based on the linear F-V relationship. The first aim of this study was to assess concurrent validity by comparing the parameters of the two-point model with directly measured F and P obtained using standard testing procedures. The second aim was to investigate whether the two-point model parameters could be generalized across the different functional movement tests that assess leg muscle capacity. Possible results could lead to a practical application of the simple two-point model as well as contribute to a better understanding of mechanical muscle capacities and the function of our muscular system.

## 2. Materials and Methods

### 2.1. Participants

Twelve physically active participants (female physical education students; age 21 ± 2 years, body mass 67.4 ± 6.2 kg, height 172 ± 7 cm) were recruited for the study. The sample sizes ranging from 3 to 12 appeared to be necessary to detect differences between dependent variables obtained from different loading conditions [13,18]. Participants reported no recent injuries or chronic diseases that could affect the performance tested. All participants were physically active during their academic curriculum, which typically included about 10 h per week of moderate physical activity, and none of them were active athletes. They did however, have experience working out in the gym. The study was conducted in accordance with the Declaration of Helsinki and all participants signed an informed consent form approved by the University of Belgrade, Faculty of Sports Review Board (ID 02-35-1).

### 2.2. Testing Procedures

Body height and body mass were measured with a standard anthropometer (Martin Anthropometer GPM 101, Duebendorf, Switzerland) and a digital scale (SECA 888 Digital Scale, SECA, Hamburg, Germany). The main part of testing procedure consisted of three functional tests for maximum performance of the leg muscles, which were carried out under different loads: countermovement jumps (JUMP), maximal cycling sprint (CYCLING) and maximal F under isokinetic conditions of the knee extensors (ISOKINETIC).

The experimental procedure used for both groups of participants was performed during the 4 sessions separated by at least three days of rest. The first test session consisted of anthropometric measurements, followed by a familiarization with JUMP, CYCLING and ISOKINETIC tests. In the second, third and fourth testing sessions, each test was performed separately. Note that the order of the tests was randomized for each participant. Moreover, the loads within each test were randomized. The sessions usually lasted about 90 min. For all tests except CYCLING, the first trial served as a practical test, while the second trial was used for further analysis. Prior to each session, each participant was given a 5-min warm-up period on a stationary bicycle, followed by 5 min of active and passive stretching exercises. Afterwards, participants had a specific warm-up consisting of several trials of jumping and isokinetic extension, but not cycling (because they already had it in the general warm-up). All measurements were performed in the university research laboratory. The process of data collection is shown in Figure 1.

### 2.3. Standard Lower Limb Tests

The test JUMP with weighted vest and belt (MiR Vest Inc; San Jose, CA, USA; weight approx. 1 kg) was performed on a force plate (AMTI, BP600400; Watertown, MA, USA). Participants were instructed to perform unconstrained maximum vertical jumps “as high as they can” from an upright, standing position with hands on hips [15]. No specific instructions were given regarding the depth of counter-movement. 

The CYCLING test included the evaluation of the maximum power output of the 6-s maximum wheel sprint [15,36,37] performed on a Monark 894E leg bike ergometer (Monark, Varberg, Sweden). Participants were instructed to perform an “all-out” effort from the beginning of the test and to remain seated throughout the sprint [15,37]. The test started with the preferred leg in the crank position at 45° forward. The seat height was adjusted for each participant based on the height of the greater trochanter while standing parallel to the seat and following the instructions of the bike ergometer [38]. 

The ISOKINETIC test was performed on the isokinetic dynamometer Kin-Com AP125 (Chatex Corp., Chattanooga, TN, USA). The participants sat in an upright position and were fastened to the test device with the straps around the pelvis, thigh and ankle. The axis of rotation of the dynamometer was aligned with the lateral femoral condyle. For the ISOKINETIC knee extension tests, the range of motion of the knee extension was set from 90° to 170° [39].

For the evaluation of the maximum F, V and P (Fmax, Vmax and Pmax, respectively) in various functional tests, the external load condition that is usually used in standard test procedures was selected. The test JUMP was carried out with unloaded vest and belt. For the test CYCLING the external load of 6 kg was used, which corresponded to approximately 8.9% of the participant’s body weight. For the assessment of force in the ISOKINETIC test, the angular velocity was 60°/s, while the angular velocity for maximum power was 180°/s [40].

### 2.4. Two-Point Model

The two-point model consisted of two loads or two velocities, depending on the test, to obtain the parameters of maximum F, V and P (F0, V0 and P0 respectively). Magnitudes corresponded to the lowest and highest loads/velocities that were used in our previous studies [8,10,15]. 

For the JUMP test, the participants performed 4 countermovement jumps (2 loads × 2 tests). The first load was performed with empty vest and belt, while the second was performed with a load of 24 kg. The trial with the highest peak P was used for further analysis. The familiarization procedure showed that all participants were able to jump with the heaviest load (24 kg). The rest period between two consecutive jumps was 1 min and 3 min between different loading magnitudes [15].

For the test CYCLING the participants performed two sprints with the lowest external load of 2 kg and with the heaviest load of 10 kg (2 loads × 1 trial). The rest period between the consecutive sprints was 4 min [15].

For ISOKINETIC, the two-point model was not applied to the lowest and highest V that the participants could perform, but rather on the most frequently used test V—60 and 180°/s (2 × 2 trials). Each trial consisted of a single contraction performed as hard as possible and the trial with the highest peak F was used for further analysis. The rests were 30 s between the trials and 1 min between 2 consecutive velocities. A real time visual feedback of the F-time curve was available during the strength assessment [22,41].

The same experienced examiner supervised all the tests. Before each test, a detailed explanation and qualified demonstration was given and a standardized verbal stimulus was given. Participants were asked to complete two to three submaximal exercise repetitions before each test series.

### 2.5. Data Analysis

With regard to JUMP, a specially developed LabVIEW program (National Instruments 2013; Austin, TX, USA) was used to record and process the vertical component of the reaction force. The signals were sampled at 1000 Hz and low-pass filtered with a second-order recursive 10 Hz low- pass Butterworth filter. Integration of the acceleration signal obtained from F was conducted to calculate V [13,42]. The analyzed motion phase covered the time interval from the lowest position of the body center of gravity to the beginning of the flight phase. Thereafter, the maximum value of F, V and P, were obtained from the jumps’ concentric phase.

Regarding CYCLING, device software (Monark anaerobic test) was used to acquire P and the frequency data. To obtain the corresponding linear measures, V was calculated from the frequency and the crank length, while F was calculated as P divided by V [15]. The maximal values were obtained for further analysis.

With regard to ISOKINETIC, a customer-specific LabVIEW program was used for data acquisition and processing. The force-time curves were recorded at 500 Hz and low-pass filtered (5 Hz) with a second-order Butterworth filter (zero phase delay). Since F was recorded directly, the angular V (rad/s) was transformed into a linear V (m/s) by multiplication with the length of individual lever arms, so the results could be comparable with other tests. The maximal values of F and V were obtained for further analyses.

### 2.6. Statistical Analysis

Descriptive statistics were calculated and the data were presented as mean and standard deviation. Prior to the statistical analyzes, initial tests showed that none of the dependent variables deviated significantly from their normal distribution (Shapiro-Wilk test). The variables: Fmax, Vmax and Pmax were assessed using standard test procedures. The two-point parameters F0, V0 and P0 were calculated by fitting a linear regression through the maximum values of the F and V data obtained from 2 loads, i.e., angular velocities, depending on a test. The F-V relationships were extrapolated to determine the maximum F (F0; F-intercept) and the maximum V (V0; V-intercept) and the slope of the relationship (a = F0/V0). Finally, the maximum P was calculated from the product of F0 and V0 (P0 = F0 × V0/4). The relationship between two-point parameters and maximum values from standard testing was tested using Pearson correlations. The Student’s *t* test for dependent samples was used to test the differences between the two-point parameters and the maximum values obtained from standard tests. The Pearson correlations and the corresponding 95% confidence intervals (95% CI) were calculated to test the relationships between the same variables between different tests. The data were analyzed using SPSS 20.0 software (SPSS Inc. Chicago, IL, USA). Alpha was set at 0.05.

## 3. Results

Figure 2 shows a two-point model of three different functional tests for the leg muscles. Two-point parameters were determined from the 1 kg and 24 kg for JUMP, 2 kg and 10 kg for CYCLING and 60 and 180°/s for ISOCINETIC. The F0 and V0 were highest for JUMP and lowest for ISOKINETIC. The steepness of the slope, which represents the ratio of F and V, was again highest at JUMP, while it was lowest at CYCLING.

Figure 3 shows the differences between the magnitudes of the same variables observed with the two-point model and standard testing procedures. The results in Figure 2 showed significant differences (*p* < 0.05) for all three tests. The highest values of F and P were evaluated in the JUMP test, while the lowest values were obtained in ISOKINETIC. Note that all values from the two-point model were higher than the values from the standard tests. Figure 2 also shows the relationship between the same variables from different tests. All correlation coefficients were found to be strong (r ≥ 0.84; *p* < 0.01), except F in CYCLING where it was low and negligible (r = −0.24).

Table 1 shows the generalizability of two-point parameters and maximum values from standard tests by correlating the same variables obtained from three different tests. In general, the correlation coefficients ranged from moderate to strong. In particular, the relationships were stronger between P (0.68 on average) than between F variables (0.47 on average) and somewhat stronger between standard tests (0.64 on average) and two-point parameters (0.51 on average).

## 4. Discussion

In this study, we investigated the parameters obtained from the two-point model in various functional tests of the lower limb tests. As for our first aim, we compared the parameters with directly measured mechanical muscle capacities assessed by standard testing procedures. The results showed that the two-point parameters were higher than the directly measured variables in all tests. The correlation between them was strong, except for the F parameter in CYCLING, which was low and insignificant. Our second aim was to determine to what extent the parameters of the two-point model can be generalized across the different tests of the lower limb tests. The results showed that the correlation coefficients between the same two-point parameters of different lower limb extremities tests ranged from moderate to strong. In particular, the relationships between P-variables were stronger than between F-variables and somewhat stronger between standard tests and two-point parameters.

Although recent studies suggested that F-V relationships could be used in routine testing [15,23], only a few of them investigated the two-point model (i.e., the load). The results of the studies mentioned above showed that the parameters obtained from the two-point model were very similar to those obtained from the linear F-V relationship. Furthermore, the investigation of [34] showed that reliability and validity were highest when the most distant pair of loads (i.e., 20% and 70% of 1 RM) was used among all two-point methods evaluated. Based on this fact, and in line with our previous study [15], we have applied the specific magnitudes of load (described under Methods) in the present study. The results showed that the two-point parameters for JUMP, CYCLING and ISOKINETIC were higher than the directly measured variables force and power. Moreover, the correlation between them was strong, except for the correlation between the parameters F0 and Fmax in CYCLING, which was low and negligible. The possible reason for this result could be that a standard test procedure involving a load of 6 kg (corresponding to 8.9% of the body mass of the participants) was the optimal load for the development of maximum power [17,37] rather than the maximum force. Note that velocity variable was excluded as it could be considered constant in ISOKINETICS. It should be noted that the distance between the applied loads was the furthest in the ISOKINETIC compared to the other two tests (see Figure 1). This could explain the highest correlation between two-point parameters and directly measured variables of F (r = 0.99) and P (r = 0.98) in this test.

In routine testing there is assumed that results obtained with only a few muscles can be partially generalized to the entire muscle system [33,35]. Our results regarding correlations between standard leg tests support these findings. In particular, P could be generalized between different leg tests, while F could only be partially generalized. To our knowledge, the relationships between muscle capacities determined by linear F-V parameters (F0, V0, P0) from different tests have so far only been presented by Zivkovic with colleagues [15]. The authors have shown that the generalization of parameters obtained from the standard regression model was inconsistent for arm and leg muscle tests. In general, the results showed that the correlation between the P-variables was higher than between the F-variables. It was concluded that the observed parameters can only be partially generalized. Similar results concerning the leg muscle tests were obtained in the current study, only between two-point parameters. In particular, a moderate correlation was observed between JUMP and CYCLING for P0, while a high degree of agreement was found between ISOKINETIC and the other two tests. These results could be explained by the fact that ISOKINETIC is considered a routine test for assessing muscle capacity [10,22,43]. Similar to the F in standard tests, the parameter F0 could only be partially generalized. These findings represent an advance in the assessment of the mechanical properties of muscles. Thus, with new methods, the mechanical properties of muscles can be assessed by only one test and thus generalized to the entire muscular system. In our case, the possibility of such generalization is much higher when it comes to P, not F, which can only be partially generalized and must be verified with multiple tests.

### Limitations

We recognize possible limitations in this article: (i) during testing procedure, the two most commonly used angular velocities (i.e., 60°/s and 180°/s) that are far from the velocity section, therefore it is possible that the accuracy of the F-V relationship could be improved by velocities closer to the velocity section by reducing the extrapolation required to achieve V0 [44]; (ii) there is some information due to the limitations of isokinetic devices for testing very fast movements [5]; (iii) only women were included in the present study, so we cannot generalize the results to both sexes; (iv) only physically active population was included in the study, so we cannot generalize the results of the study to the non-active population; (v) the sample size is rather small, but in line with previous studies in the same field [10,13]; (vi) we did not control for the depth of the squat within the countermovement jumps, however, majority studies from this specific field did not control for the depth of the squat when determining F-V parameters with novel two-point method [13,15,42,45]; furthermore, to minimize the possible effect of the depth of the squat we gave participants instruction “jump as high as you can”; (vii) the nature of the countermovement jump provides an approximate small distance between loads; nevertheless, the F-V parameters in the countermovement jump test have been shown to be reliable and valid [13,14,22,45,46].

## 5. Conclusions

In summary, the present study showed a high degree of agreement between standard tests and the novel two-point model in general. Furthermore, the results showed that the mechanical capacities of the leg muscles can be partially generalized between different functional tests. The addition of only one additional load or velocity to the standard functional tests of muscle capacities could distinguish the basic mechanical capacities of the tested muscles. A fairly consistent data set observed when comparing maximum power from standard tests and P0 from the two-point model suggests that it could be used for routine testing. The two-point model could further improve test protocols by allowing easier and faster assessment of maximum F, V and P. Although the correlations for the same variables obtained from different standard tests were moderate to high, further investigation is needed. Further investigation should include more different functional tests performed on different types of subject samples to assess the validity and sensitivity of two-point parameters in the future. In addition, for the application of the two-point model in practice, the methodology must first be standardized, which includes the selection of the type and magnitudes of load and velocity included in testing procedures.

## Figures and Tables

**Figure 1 ijerph-18-01032-f001:**
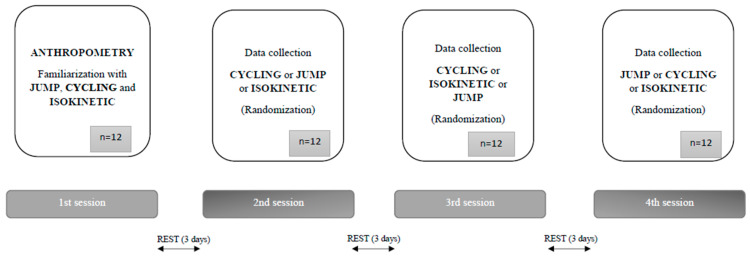
Flow chart of data collection.

**Figure 2 ijerph-18-01032-f002:**
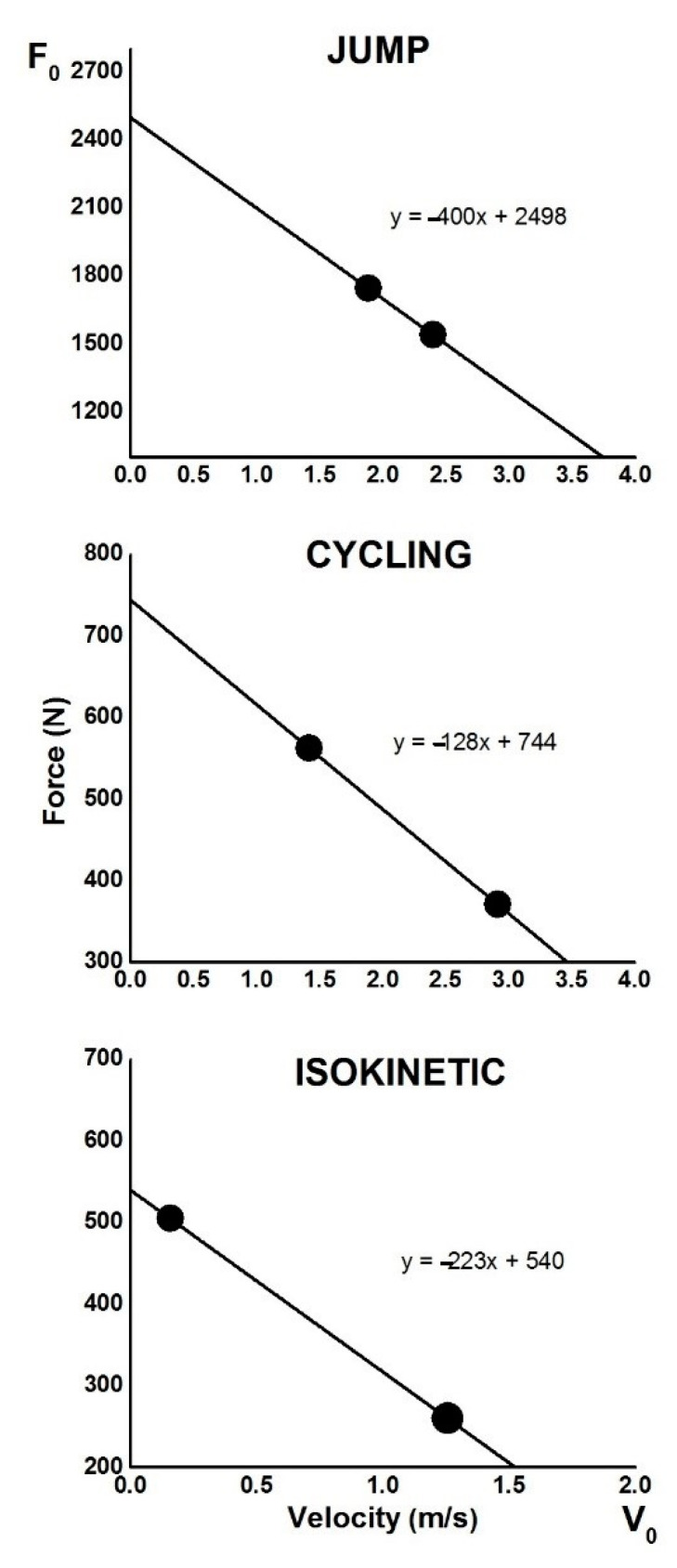
Two-point model presented for three different functional tests of lower limb muscles averaged across participants.

**Figure 3 ijerph-18-01032-f003:**
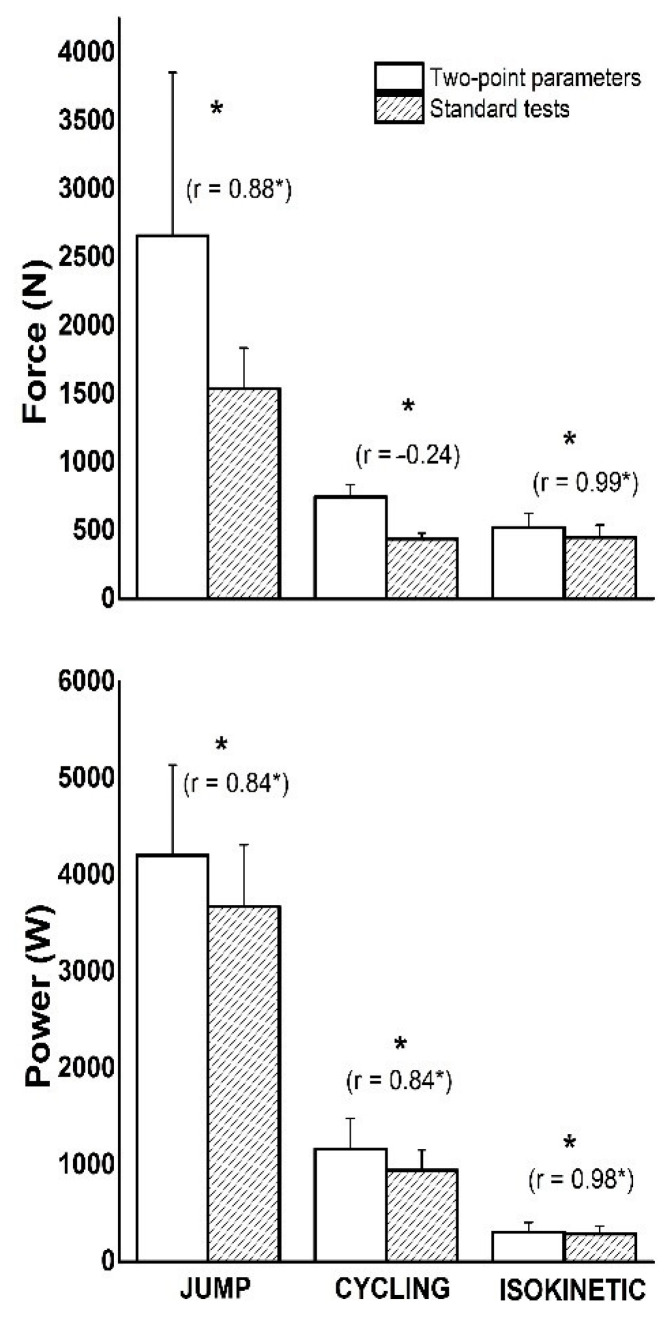
The averaged across the participant values of F (top panel), and P (bottom panel) obtained from the two-point model (open bars) and standard tests (filled bars) for all lower limb tests (means with SD error bars). The correlation coefficients are presented in parentheses above bars (* *p* < 0.05—significance of correlation). Significant differences between the two-point parameters and maximal values obtained from standard tests are marked (* *p* < 0.05).

**Table 1 ijerph-18-01032-t001:** Pearson’s correlation coefficients observed among the same two-point parameters and maximal values obtained from standard tests between three different leg tests.

		Standard Tests	Two-Point Method
**F**	JUMP-CYCLING	0.55 (−0.04–0.85)	0.49 (−0.12–0.83)
	JUMP-ISOKINETIC	0.49 (−0.12–0.83)	0.23 (−0.39–0.71)
	CYCLING-ISOKINETIC	0.57 (−0.01–0.86)	−0.47 (−0.14–0.82)
**P**	JUMP-CYCLING	0.66 * (0.14–0.89)	0.49 (−0.12–0.83)
	JUMP-ISOKINETIC	0.78 ** (0.37–0.94)	0.72 ** (0.25–0.92)
	CYCLING-ISOKINETIC	0.77 ** (0.35–0.93)	0.66 * (0.14–0.89)

In parentheses are shown 95% CI for corresponding correlation coefficient (* *p* < 0.05; ** *p* < 0.01—significance of correlations).

## Data Availability

Data generated and analyzed during this study are included in this article. Additional data are available from the corresponding author on request.

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
