# Peer review of "Are the Parameters of Novel Two-Point Force-Velocity Model Generalizable in Leg Muscles?"

_ijerph, 2021, doi:10.3390/ijerph18031032_

Round 1

Reviewer 1 Report

Firstly, I want to thank the editor for the opportunity to review the manuscript "Are the Parameters of Novel Two-Point Force-2 Velocity Model Generalizable in Leg Muscles?". Tha paper looks particularly interesting as it discuss relevant methodological issues in testing skeletal muscle parameters.

In my opinion the paper is well written and the results are support by adequate methodological choices, as the testing protocol and statistical analysis process. Additionally, the limitations paragraph clearly present possible issues that should be addressed in future studies.

I have only a minor suggestion, since the readership of this journal: testing muscle parameters is one of the outcomes in different environmental scenarios and physiological conditions to understand adaptations and responses. A paragraph discussing such factors as temperature (e.g., Todd et al., J Physiol, 2005; Racinais & Oksa, Scand J Med Sci Sports, 2010; Morrison, Sleivert & Cheung, Eur J Appl Physiol, 2004), altitude (e.g., Morales-Artacho et al., Front Physiol, 2018; Garcia-Ramos et al., J Strenght Cond Res, 2018), or dehydration (e.g., Pallarés et al., J Int Soc Sport Nutr, 2016; Judelson et al., Sports Med, 2007; Zubac et al., Eur J Sports Sci, 2020), may help commenting the usefulness of the proposed new model in environmental research.

Reviewer 2 Report

The authors present a study in which they test the efficacy of the two-point force velocity model.  In general the study and manuscript has merit.  However, the manuscript needs significant attention regard its use of the english language.  The are many grammatical errors and repeated words.  Please seek help with this.

From a scientific point of view, the authors need to restructure the introduction to better explain to the reader what they mean by two-point force velocity model.  Moreover, the authors need to address the fact that their two data points for the 'jump' test are much closer together that the other modes of exercise.  How does this effect the validity of the 'jump' tests and how well do these data compare to the other test modes?

Reviewer 3 Report

In this study the authors compare the two-point model with an standard test in determining the parameters linked to the linear force-velocity relationship in 3 different exercises (counter-movement jumps, maximal cycling sprint and maximal force under isokinetic conditions of the knee extensors), with the aim of improving the knowledge of the muscle mechanical capacities and contribute to the further development of lower-limb routine tests.

From my point of view, the manuscript, under the current conditions, does not present a good approach to the aims and presents large gaps in the methodology that make it difficult to further understand the process applied. I do not understand exactly why the authors want to compare Fmax, Vmax and Pmax vs F0, V0 and P0 between tests and between movements, nor how these comparisons or associations can improve the current knowledge about muscle mechanical capacity. The authors need to organize their thoughts and bridge the gap between ideas better, so that the originality and objective of his work is evident. The lack of control in some of the movements studied (for example, the depth of the jump), joined to an small sample that no domain of the exercise (active women with a single familiarization session), and others  reasons described by the authors in the limitations section, makes the results obtained can be unreliable. Within-subject reproducibility of the results is not provided in the manuscript. The statistics test applied present some mistakes in some sections, and the statistic design can be notably improved.

I am aware of the efforts of the researchers in this work, so I hope that these comments do not discourage them and work to improve the approach and the presentation of their results.  

Specific comments

Abstract: In the way the abstract is written, it is very difficult to understand the purpose and procedure used in this study.

-  lines 12-13: What do the authors mean by directly measuring P?

- line:14-17. Methodology displayed is confuse

 - line 18:  “The results showed significant differences (p <0.05; paired t-test) for all three tests”. It does not provide information on the direction of the results or the magnitude of de change. Please delete the sentence and present the main findings of your research.

- line 21. What is a standard test?

- Line 22-23: Do you refer to the same variable between different exercises? Or between the same variable and the two tests? I imagine that when I go further in the manuscript, I will be able to understand a little more about these results. In any case, as it is written it seems the second option. I do not understand how these results lead you to your conclusions. The standard procedure that it seems to apply does not allow determining a F-V profile but rather an isolated data of F, V or P, so both tests report different partial aspects of the muscle mechanic capacity.

- Please remove the second part of your conclusion from this section 

Introduction: The authors need to organize their thoughts and bridge the gap between ideas better, so that the originality and objective of his work is evident.

Line 39-40. What do the authors refer to as traditional systems? These types of models do not correspond to employees in this field at least in the last decade. 

Line 40. Eliminate the repeated word.

Line 66-67: what do you mean with “standard test procedures”? to the traditional procedures referred at the beginning of the paragraph?

Line 71-72: Please indicate what exactly the meaning of “there are no results to generalize the F-V relationship between the results of different functional measurements”

Method: This section present large gaps that make difficult to further understand the process applied and reproduce the study. The exact procedure and test used must been fully described.

- 12 physical active women participated in the study

- The authors indicate that the participants report an average of 10 hours/week of moderate physical activity. However, the action included in the study and the intensity required (maximal) require technical mastery and strength to ensure a reproducibility of the results in each condition and movement. A familiarization session to the JUMP, CYCLING and ISOKINETIC is not enough to save this inconvenience.

- Was the order of the tests in terms of the “type of movement” randomized? Was the type of test randomized for each exercise? or were both tests performed in the same order in all cases?

- Did the authors included specific warming-up for each movement?

- What rest time between tests of the same day was established?

- line 114-115. Which trial?

- Lines 122-125-Why did not you measure the depth of the jump?  

- Lines 138-140: How do the authors select the external load considered as "usual" in the standard evaluation of the applied tests?

- Lines 146: please, define F0, V0 and P0

- Line 150: Why did the authors select 24 kg for all participants as heavy load? The results show two near points in the resulting FV profile.

-How did the authors verify that the participants performed the CMJ with the same depth with and without overload?

- Was the unloaded jump in the 2-point test considered as the “standard test”? or, were they both performed separately?. In this case, the reliability between the standard test and the unloaded jump of the 2-point test can be checked to determine the quality of the study sample.

- Was the 2-point test performed on the same force platform?

- Line 164: please specify the load per repetition, the rest between loads, trials and between the last trial and the beginning of the test.

Statistics

- For the sample size used, the Kolmogorov test is not adequate to determine the normality of the study variables.

- Line 184: The Fmax, Vmax y Pmax have not been described in the method section.

- Lines 190-192: I do not fully understand the purpose of this test given the difference in the magnitude of the variables. Obviously, an F max value in an unloaded CMJ is must be much lower than an F0 value in a 2-point test. It is understood that you do not compare the same variable between the different movements, however, the abstract provides information in this regard. Perhaps, if authors are interested in performing comparative analyzes between the F values of both types of tests, they should have used other test such as factorial ANOVA with 2 repeated factors: type of movement and type of test.

- Perhaps it could be interesting to apply multiple regression models to determine if the prediction models of the F or the P of the movements analyzed explain different components of the variance given the low correlation between them in most of the cases shown in Table 1.

Results

- the distance between the two jump test points in fig 2 are too close, so the estimation of F0 is likely to give a large error

- Line 201: 0 kg or 1 kg?

- Fig 3: Please report your F and P data as relative body weight values rather than absolute values.

- line 214: please eliminate paired t-test.

- Please provide the exact p-value and IC better than an asterisk

- line 223-227: the correlations in general are weak-moderate and fairly strong in only 3 cases. Rather than discuss whether the strength of the peer association changes, throughout the discussion the authors should discuss the nature of the association and its application.

No data is provided in tables, figures or writing on the d of Cohen indicated in the statistics section.

Round 2

Reviewer 3 Report

I am grateful to the authors for the revision effort they have made of their manuscript, however, the methodological deficiencies that led me to reject the manuscript cannot be corrected and remain in the current version of the manuscript. I do not agree with the authors that a "physically active" sample enables the ability to adequately perform an overloaded or free jump test. Countermovement and overloaded jumps are not a common activity in a gym. Authors refers the study of Mandic et al, 2015 to asseverate the reliability of the self-selection depth. However, the sample used by Mandic et al, 2015, are professional male basketball players, a quite different population from the one used in this study. A marked forward flexion of the trunk or a change in knee angulation between heavy or light loads could have happened. In order to ensure a valid result, it is necessary that the technique does not affect the variables measured but, in this case, technical capacity and reliability of the jumps have not been controlled.

On the other hand, the two points of the 2-load test in the vertical jump are remarkably close, so the error in the extrapolation of the theoretical maximum values (F0 and V0) is sure to be large. The error of the obtained F–V relationship (and, therefore, of F0, V0, and P0max) would be smaller if calculated from more distant experimental points (Jaric, 2016). 

In my opinion, the validity of the data obtained from the vertical jump test cannot be assured, which calls into question the conclusions of the study.
